# Dendritic Integration Inspired Artificial Neural Networks Capture Data Correlation

Chongming Liu[a,b], Jingyang Ma[a,b], Songting Li[a,b,c,1], and Douglas Zhou[a,b,c,d,1]

[a]School of Mathematical Sciences, Shanghai Jiao Tong University, Shanghai, China
[b]Institute of Natural Sciences, Shanghai Jiao Tong University, Shanghai, China
[c]Ministry of Education Key Laboratory of Scientific and Engineering Computing, Shanghai Jiao Tong University, Shanghai, China
[d]Shanghai Frontier Science Center of Modern Analysis, Shanghai Jiao Tong University, Shanghai, China

## Abstract

Incorporating biological neuronal properties into Artificial Neural Networks (ANNs) to enhance computational capabilities is under active investigation in the field of deep learning. Inspired by recent findings indicating that dendrites adhere to quadratic integration rule for synaptic inputs, this study explores the computational benefits of quadratic neurons. We theoretically demonstrate that quadratic neurons inherently capture correlation within structured data, a feature that grants them superior generalization abilities over traditional neurons. This is substantiated by few-shot learning experiments. Furthermore, we integrate the quadratic rule into Convolutional Neural Networks (CNNs) using a biologically plausible approach, resulting in innovative architectures—Dendritic integration inspired CNNs (Dit-CNNs). Our Dit-CNNs compete favorably with state-of-the-art models across multiple classification benchmarks, e.g., ImageNet-1K, while retaining the simplicity and efficiency of traditional CNNs. All source code are available at `https://github.com/liuchongming1999/Dendritic-integration-inspired-CNN-NeurIPS-2024`.

## 1 Introduction

While the Artificial Neural Network (ANN) framework has made significant advancements towards solving complex tasks, it still faces problems that are rudimentary to real brains [6]. A notable distinction between the modern ANN framework and the human brain is that the former relies on a significant number of training samples, which consumes large amounts of energy, whereas the latter runs on extremely low power (<20 watts) and possesses a strong generalization capability based on few-shot learning. Studies have demonstrated that incorporating dendritic features in ANNs can alleviate these issues and enhance overall performance [52, 38, 30]. However, it is difficult to quantify the nonlinear integration of dendrites, which is an essential property that allows individual neurons to perform complex computations [43, 49]. As a result, dendritic-inspired models often employ linear integration with nonlinear activation functions such as ReLU and Sigmoid [21, 37].

In light of recent studies revealing that the somatic responses of biological neurons to multiple synaptic inputs on dendrites follow a quadratic integration rule [15, 28], we explore the computational benefit of the quadratic neuron model [41]. This model substitutes the linear integration and nonlinear

---

[1]Corresponding author, songting@stju.edu.cn, or zdz@sjtu.edu.cn

activation function of traditional point neurons with quadratic integration as follows[1]:

$$f(x) = \sigma(w \cdot x + b) \rightarrow f(x) = x^T A x + w \cdot x + b. \tag{1}$$

The expected value of a quadratic term $E_{x \sim \mathcal{D}}[a_{ij} x_i x_j]$ closely relates to the covariance between variables $x_i$ and $x_j$ ($x_i$ and $x_j$ are the $i$-th and the $j$-th components of vector $x$, respectively), suggesting an inherent capability for capturing input correlation. This attribute allows biological neurons to exhibit remarkable performance in correlation detection tasks [1, 29]. Here, we theoretically demonstrate that quadratic neurons indeed naturally capture correlation within structured data, which is pivotal in numerous machine learning applications [7], such as language generation [5] and video understanding [23]. This intrinsic property endows quadratic neurons with superior generalization capabilities compared to traditional neurons, as evidenced by few-shot learning experiments. We further propose a biologically plausible method to integrate them into Convolutional Neural Networks (CNNs). Applying this approach to ResNet [16] and ConvNeXt [34] results in novel CNN models—Dendritic integration inspired ResNet (Dit-ResNet) and Dendritic integration inspired ConvNeXt (Dit-ConvNeXt), respectively. Evaluations on CIFAR-10 and CIFAR-100 datasets demonstrate that Dit-ResNets significantly enhance test accuracy by merely replacing a single layer of point neurons with quadratic neurons. On the ImageNet-1K dataset [10], Dit-ConvNeXts demonstrate significant enhancements in top-1 accuracy with only a one percent increase in parameters, exhibiting competitive performance compared to state-of-the-art models. Ablation studies further substantiate the effectiveness of quadratic neurons and their capability for capturing data correlation.

This paper is structured as follows: Section 2 reviews previous works related to dendritic-inspired models and high-order interactive operations in neural networks. Section 3 presents a theoretical analysis of the computational benefits derived from quadratic neurons, as demonstrated through few-shot learning experiments. Section 4 describes the architecture of Dit-CNNs and evaluates their performance on several computer vision benchmark datasets. Section 5 concludes the paper. All numerical experiments in this paper are conducted using Python and executed on 8 Tesla A100 computing cards with a 7nm GA100 GPU, featuring 6,912 CUDA cores and 432 tensor cores.

## 2 Related work

**Dendritic-inspired computational models.**  The significant role of dendrites in the nonlinear integration within neural systems is well-recognized, enabling neurons to perform intricate tasks [43, 49, 2]. Recently, numerous studies have explored the implementation of dendritic features in ANNs from different aspects, yielding encouraging results. In [52, 21, 37], the integration of dendritic morphology into ANNs has led to higher test accuracies on simple tasks compared to traditional ANNs. Furthermore, dendritic plasticity rules have been employed to develop learning algorithms in [38, 36], effectively replacing the non-biological backpropagation algorithm and achieving enhanced performance on classification tasks with small data sizes. Additionally, in [30], dendritic nonlinearity is considered by adding input to each layer with the Hadamard product rule, which has shown improvements in approximation and classification experiments.

**Neural networks with high-order interactions.**  Recent advancements in the design of deep learning architectures have been predominantly driven by the ability to capture high-order statistics among inputs and features. Over recent decades, there has been a pivotal shift in the foundational neural networks used for computer vision tasks: moving from convolution-based architectures [16, 24, 56] to Transformer-based models [11, 33, 9]. This transition has been driven by the development of the self-attention mechanism, which facilitates quadratic interactions among inputs. Additionally, a novel category of neural networks, referred to as Mamba [14], has demonstrated exceptional performance in tasks that require the processing of extensive high-order statistical information, such as video understanding [32, 59, 27]. The backbone of Mamba's architecture is a state space model that enables high-order interactions among inputs. Moreover, the integration of high-order spatial interactions into CNNs has also been shown to enhance performance significantly [40]. These developments underscore the significance of high-order statistical information in tackling large-scale complex tasks and suggest that incorporating high-order interactions as a foundational aspect of model design is an effective strategy.

---

[1]$A$ can always be considered a symmetric matrix, attributable to the quadratic form present in $f(x)$.

Table 1: The overview of current works with quadratic format.

| Reference | Quadratic format | How quadratic operation is used | Biological interpretation | Theory for generalization |
|---|---|---|---|---|
| [54],[12],[4], [53] | $f(x) = (w_a x)(w_b x)$ | Pixel-Wise | N | N |
| [48],[20],[35],[60] | $f(x) = x^T A x$ | Pixel-Wise | N | N |
| Dit-CNNs | $f(x) = x^T A x$ | Channel-Wise | Y | Y |

Previous works have explored the concept of quadratic integration as a sophisticated alternative to traditional linear summation. Table 1 summarizes various neuron models formulated in quadratic formats, along with their relevant references. The one-rank format of quadratic neurons, when applied across entire networks, is documented in [54, 12, 4, 53], demonstrating enhanced performance in classification tasks. Studies such as [48, 20, 35, 60] have implemented quadratic neurons on a pixel-wise basis within convolution layers, yielding modest improvements in test accuracy. In contrast, Table 1 illustrates how our Dit-CNNs approach departs from these models by integrating a channel-wise quadratic operations with biological interpretation and providing a theoretical analysis of the computational advantages offered by quadratic neurons.

## 3 Quadratic neurons possess enhanced generalization capabilities

This section provides a theoretical analysis demonstrating how quadratic neurons enhance generalization capabilities by effectively capturing correlation within structured data. Additional numerical experiments corroborate this assertion. The detailed proofs can be found in Appendix A.

### 3.1 Binary classification for normal distributions

Assume that the data points of two classes are equally sampled from two different non-degenerate normal distributions: $class_1 \sim N(\mu_1, \Sigma_1)$, $class_2 \sim N(\mu_2, \Sigma_2)$, where $\mu_j \in \mathbb{R}^d$, $\Sigma_j \in \mathbb{R}^{d \times d}$ ($j = 1, 2$). Then the optimal classifier $y_{opt}(x) : \mathbb{R}^d \to \{1, 2\}$ can be defined according to the sampling probability:

$$y_{opt}(x) = \arg\max_{j \in \{1,2\}} p_j(x), \tag{2}$$

where $p_j(x)$ is the probability (density function) of sampling point $x$ from distribution $N(\mu_j, \Sigma_j)$.

If a single quadratic neuron, as described in Equation (1) ($f(x) = x^T A x + w \cdot x + b$, where $A$ is a symmetric matrix), is used to solve the above binary classification task, we can prove the following theorems:

**Theorem 1.** *(Existence) The critical points with respect to the cross-entropy loss $L(A, w, b)$ are given as follows:*

$$A^* = \Sigma_1^{-1} - \Sigma_2^{-1}, \ w^* = -2(\Sigma_1^{-1}\mu_1 - \Sigma_2^{-1}\mu_2), \ b^* = \mu_1^T \Sigma_1^{-1}\mu_1 - \mu_2^T \Sigma_2^{-1}\mu_2 + \log\left(\frac{|\Sigma_1|}{|\Sigma_2|}\right).$$

*i.e.*

$$\frac{\partial L}{\partial A}\Big|_{A^*, w^*, b^*} = 0, \quad \frac{\partial L}{\partial w}\Big|_{A^*, w^*, b^*} = 0, \quad \frac{\partial L}{\partial b}\Big|_{A^*, w^*, b^*} = 0,$$

*where*

$$L(A, w, b) = \frac{1}{2}\left[E_{x \sim class_1}\left(\log(1 + e^{f(x)})\right) + E_{x \sim class_2}\left(\log(1 + e^{-f(x)})\right)\right].$$

*Moreover, the corresponding classifier generated by this formula is the same as the theoretically optimal classifier in Equation* (2).

**Theorem 2.** *(Uniqueness) Consider the conditional cross-entropy loss defined on a set $\Omega \in \mathcal{M}(\mathbb{R}^d)$, where $\mathcal{M}(\mathbb{R}^d)$ denotes the Lebesgue $\sigma$-algebra on $\mathbb{R}^d$:*

$$L(A, w, b \mid \Omega) = \frac{1}{2}\left[E_{x \sim class_1, x \in \Omega}\left(\log(1 + e^{f(x)})\right) + E_{x \sim class_2, x \in \Omega}\left(\log(1 + e^{-f(x)})\right)\right],$$

*where*

$$E_{x \sim class_j, x \in \Omega}\left(g(x)\right) = \int_\Omega g(x) p_j(x)\, d\mu,$$

*then the unique solution satisfying*

$$\frac{\partial L(A, w, b \mid \Omega)}{\partial A}\bigg|_{A^*, w^*, b^*} = 0, \ \frac{\partial L(A, w, b \mid \Omega)}{\partial w}\bigg|_{A^*, w^*, b^*} = 0, \ \frac{\partial L(A, w, b \mid \Omega)}{\partial b}\bigg|_{A^*, w^*, b^*} = 0$$

*for every $\Omega \in \mathcal{M}(\mathbb{R}^d)$ is given by:*

$$A^* = \Sigma_1^{-1} - \Sigma_2^{-1}, \ w^* = -2(\Sigma_1^{-1}\mu_1 - \Sigma_2^{-1}\mu_2), \ b^* = \mu_1^T \Sigma_1^{-1}\mu_1 - \mu_2^T \Sigma_2^{-1}\mu_2 + \log\left(\frac{|\Sigma_1|}{|\Sigma_2|}\right).$$

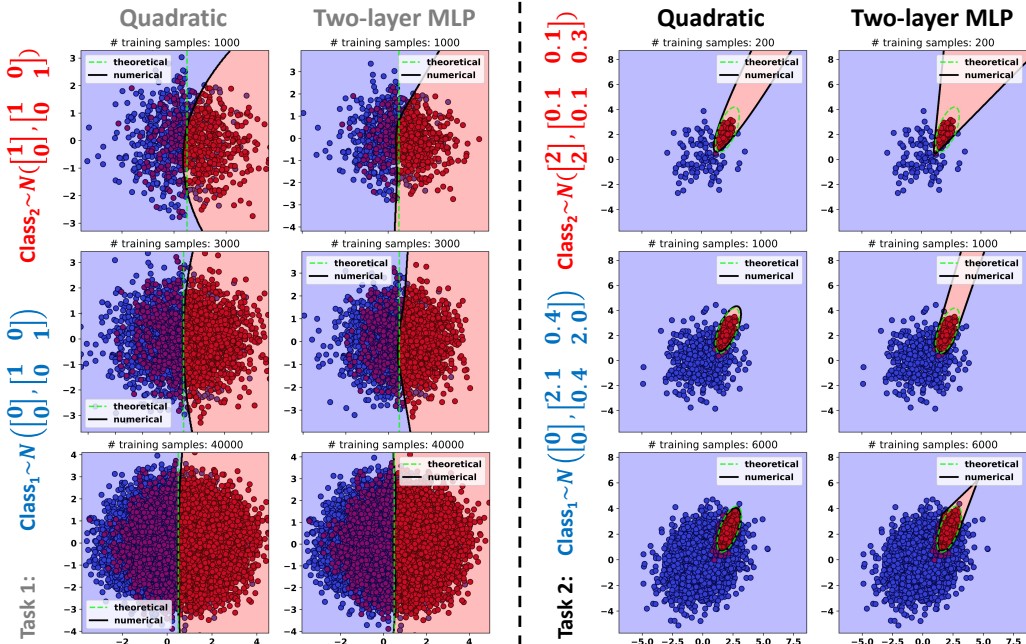

Figure 1: **Visualization of classifier boundaries for two models (a single quadratic neuron with 2-dimensional inputs and a two-layer MLP comprising 2-ReLU(10)-1) across two distinct tasks.** For each task, the impact of varying training sample sizes is examined. Model comparisons are performed under uniform conditions, utilizing identical random seeds and hyperparameters for fairness. Training is executed using gradient descent with a learning rate of 0.1 over 10,000 epochs. The term "theoretical" denotes the optimal classifier's boundary as specified in Equation (2), while "numerical" represents the empirical classification boundary obtained from the model.

Theorem 1 identifies an analytical critical point for a quadratic neuron with cross-entropy loss, which emerges as the optimal classifier. On the other hand, Theorem 2 establishes the uniqueness of this critical point under conditional cross-entropy loss, where $\Omega$ denotes the regions occupied by distinct batches of data points. Under these conditions, the quadratic parameters will converge to this unique critical point when using the stochastic gradient descent (SGD) algorithm. It is demonstrated that, unlike traditional neuron, quadratic neuron inherently possesses the ability to converge to the optimal classification solution by directly including the covariance matrix. This finding is supported by simulated data in Task 1 (identical covariance) and Task 2 (non-identical covariance) shown in Figure 1, which indicates that quadratic neuron consistently achieves the theoretically optimal classification outcome. Moreover, in comparison to a multi-layer perceptron (MLP) with two layers, quadratic neuron surpass traditional neuron in Task 2 which require capturing correlation information from data distributions (require fewer training samples to converge to the optimal solution). This highlights the unique capability of quadratic neurons to identify internal correlations within structured data, offering a possible explanation for their superior generalization capability compared to traditional neurons, a topic that will be further explored in numerical experiments below.

## 3.2 Few-shot learning experiments on MNIST and Arabic MNIST

From the above classification tasks for normal distributions as depicted in Figure 1, it is suggested that quadratic neurons may require fewer training samples in comparison to traditional neurons. We next examine this few-shot learning capability of quadratic neurons on the MNIST [25] and Arabic MNIST datasets [19]. The results, presented in Figure 2, indicate that quadratic neurons outperform MLP with two layers when trained with a limited number of samples on both datasets. This evidence shows the enhanced generalization ability of quadratic neurons in applications. Further insights into the mechanism through which quadratic neurons capture data correlation in the MNIST dataset, and a related theorem concerning multi-class classification for normal distributions, are detailed in Appendix B.

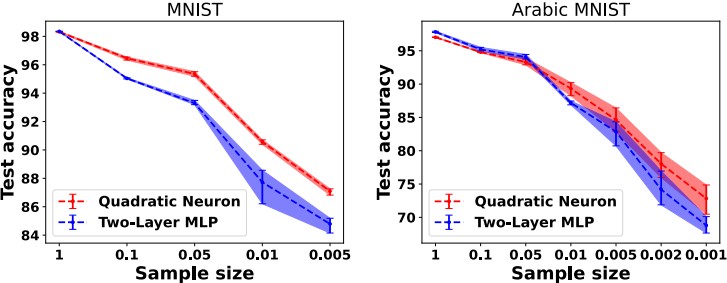

Figure 2: **Performance of two models on few-shot learning tasks with MNIST and Arabic MNIST datasets.** The first model consists of 10 quadratic neurons with a 784-dimensional input, while the second model is a MLP with the configuration of 784-ReLU(8000)-10. Both models are evaluated under identical conditions using the same training protocol, which includes SGD with a learning rate of 0.1 and a batch size of 100 across 20 epochs. The term 'Sample size' refers to the ratio of the number of training samples to the full training set. Experiments are conducted for each model and sample ratio across ten runs, and the resulting test accuracy is depicted through an error bar plot (error bar represents the upper and lower bound for test accuracy).

## 4 Integrating quadratic neurons into CNNs with biological plausibility

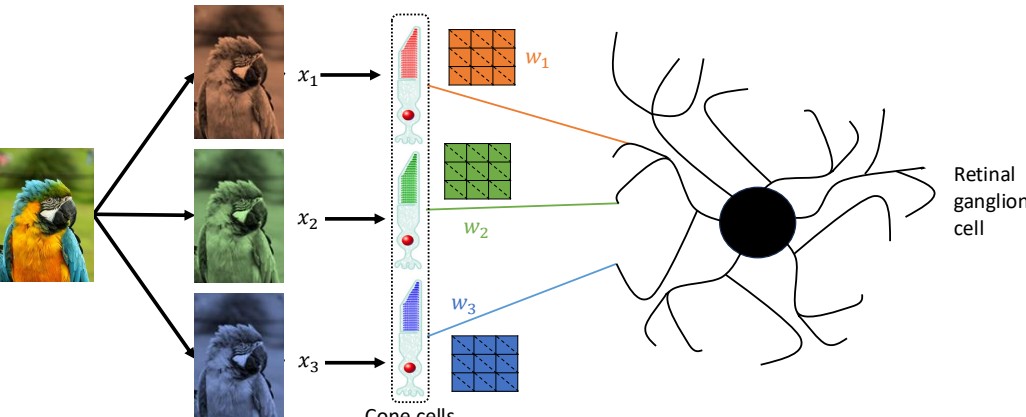

Figure 3: **Schematic of the biological interpretation of Dit-CNNs.** Dit-CNNs is inspired by neural networks in the visual system. For example, different types of cone cells encode various color (channel) information, and retinal ganglion cells receive inputs from multiple types of cone cells [22], the responses can be modeled as having receptive fields (convolutional kernels) related to different color channels ($w_1 * x_1, w_2 * x_2, w_3 * x_3$). When multiple channel inputs are present, traditional CNNs simply linearly sum the corresponding responses. In contrast, dendritic neurons integrate these inputs with an additional quadratic term based on the dendritic quadratic integration rule. This approach leads to the formulation of Dit-CNNs after simplification.

The concept of convolution [13], inspired by the discovery of the receptive field in the cat visual cortex [18], laid the foundation for CNNs [26] which have been designed to efficiently tackle large-scale vision tasks. The underlying biological interpretation of convolution is that each neuron exhibits varying responses at different image locations based on its receptive field (convolution kernel). Traditionally, CNNs have focused solely on the linear integration of inputs from presynaptic neurons. Specifically, for an input $X \in \mathbb{R}^{C_{in} \times H_{in} \times W_{in}}$, where $C_{in}$ represents the number of input layer neurons (channels), and $H_{in}$ and $W_{in}$ denote the height and width of the input feature, respectively. Given convolution kernels $w \in \mathbb{R}^{C_{in} \times C_{out} \times (2l+1) \times (2l+1)}$, the convolution output, $Y = Conv(X) \in \mathbb{R}^{C_{out} \times H_{out} \times W_{out}}$, represents the response of neuron $i$ at location $[j, k]$ to inputs from locations $[j - l : j + l, k - l : k + l]$ across all input layer neurons:

$$Y[i, j, k] = \sum_{m=1}^{C_{in}} w[m, i, :, :] * X[m, j - l : j + l, k - l : k + l]. \qquad (3)$$

However, acknowledging that neurons with dendrites obey a quadratic integration rule, we propose a novel approach that incorporates quadratic neurons into CNNs as depicted in Figure 3. This adaptation incorporates a quadratic integration term among inputs from different neurons (channels). Specifically, with a quadratic integration coefficient $A \in \mathbb{R}^{C_{out} \times C_{in} \times C_{in}}$, the output neuron's response in Equation (3) is modified as follows:

$$Y[i, j, k] = \sum_{m=1}^{C_{in}} w[m, i, :, :] * X[m, j - l : j + l, k - l : k + l] + \underbrace{X^T[:, j, k]A[i, :, :]X[:, j, k]}_{\text{Quadratic integration}}. \qquad (4)$$

### 4.1 Evaluations on CIFAR-10 and CIFAR-100

**Dataset and models.** Our initial experiments utilize the CIFAR dataset, which includes 50,000 training images and 10,000 testing images. The experimental setup follows the ResNet configurations as outlined in [16], comprising models such as ResNet-20, ResNet-32, ResNet-56 and ResNet-110. To reduce computational demands, we incorporate quadratic neurons specifically into one layer—namely, the second layer of the first block in the second stage of the ResNet architecture [16]. Additionally, to address issues related to gradient dynamics, such as gradient explosion, a Layer Normalization (LN) layer [3] is implemented immediately before the layer equipped with quadratic neurons.

Table 2: Comparative performance of Dit-ResNets and structurally similar models on CIFAR.

| Model | # Param. (CIFAR10) | Acc. (CIFAR10) | # Param. (CIFAR100) | Acc. (CIFAR100) |
|---|---|---|---|---|
| ResNet-20[16] | 0.27M | 91.25% | 0.30M | 67.26±0.68% |
| QResNet-20[12] | 0.81M | 92.22% | 0.84M | 67.82±0.52% |
| QuadraResNet-20[54] | 0.81M | 92.21% | 0.84M | 68.02±0.44% |
| Dit-ResNet-20 | 0.30M | **92.66%** | 0.33M | **68.66±0.34%** |
| ResNet-32[16] | 0.46M | 92.49% | 0.49M | 68.52±0.55% |
| QResNet-32[12] | 1.39M | 93.10% | 1.42M | 69.41±0.48% |
| QuadraResNet-32[54] | 1.39M | 93.11% | 1.42M | 69.54±0.44% |
| Dit-ResNet-32 | 0.49M | **93.17%** | 0.52M | **69.68±0.32%** |
| ResNet-56[16] | 0.86M | 93.03% | 0.89M | 70.17±0.67% |
| QResNet-56[12] | 2.55M | 93.66% | 2.58M | 71.21±0.44% |
| QuadraResNet-56[54] | 2.55M | 93.79% | 2.58M | 70.98±0.76% |
| Dit-ResNet-56 | 0.89M | **93.90%** | 0.92M | **71.40±0.35%** |
| ResNet-110[16] | 1.73M | 93.57% | 1.76M | 70.84±0.76% |
| QResNet-110[12] | 5.17M | 93.88% | 5.20M | 71.58±0.87% |
| QuadraResNet-110[54] | 5.17M | 93.77% | 5.20M | 71.72±0.81% |
| Dit-ResNet-110 | 1.76M | **94.33%** | 1.79M | **72.40±0.85%** |

**Experimental settings.** The training process incorporates SGD with a weight decay of 0.0001, a batch size of 128, and a momentum of 0.9. Following the recommendations from [12], quadratic

integration parameters are initialized to zero, and a distinct learning rate is adopted for these parameters. Initially, the learning rate for the quadratic integration matrix is set at 1, and 0.1 for all other parameters. These rates is reduced by a factor of ten at 80 and 120 epochs, with training concluding at 160 epochs. Data augmentation procedures mirror the original protocol: each image is padded by 4 pixels on each side, follows by random cropping of a $32 \times 32$ section from the padded image or its horizontal flip. For testing, a single view of the original $32 \times 32$ image is evaluated.

**Results.** Table 2 presents a comparative analysis of CIFAR performance between our Dit-ResNets and the original ResNet model [16], as well as two adaptations that incorporate quadratic neurons [12, 54], which have shown superior outcomes among existing approaches with quadratic formats [48, 20, 35, 60, 4]. For the CIFAR-10 dataset, given the extensive benchmarking in prior studies, we conduct ten runs of our Dit-ResNets, reporting the optimal outcome for comparative analysis. For CIFAR-100, we independently execute all models, documenting average test accuracy and variance (mean±std). Our experiments indicate significant performance enhancements on both CIFAR-10 and CIFAR-100 datasets after integrating quadratic neurons into a single layer of ResNet, confirming their efficacy. Furthermore, when comparing a deeper ResNet model with our Dit-ResNet (e.g., ResNet-110 vs. Dit-ResNet-56), our approach not only boosts performance but also reduces training overhead. This suggests that augmenting the model with quadratic integration parameters is more effective than merely increasing network depth—a traditionally favored strategy. In comparison to other models employing quadratic neurons [12, 54], our Dit-ResNets consistently achieve the highest test accuracy with the fewest parameters. This underscores the efficacy of our approach in leveraging the computational advantages of quadratic neurons. These neurons, as discussed in Section 3, demonstrate enhanced generalization capabilities.

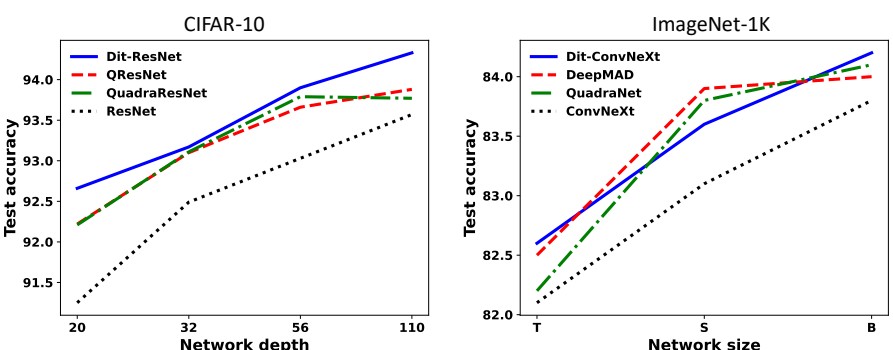

Figure 4: Visualization of some performance results presented in Table 2 (left) and Table 3 (right).

## 4.2 Evaluations on ImageNet-1K

**Dataset and settings.** We extend our investigations to the ImageNet-1K dataset [10], which comprises 1.28 million training images and 50,000 validation images across 1,000 categories. Our analysis primarily focuses on top-1 accuracy on the validation set. Training is conducted at a resolution of $224 \times 224$, supplemented by a comprehensive suite of data augmentation and regularization strategies. These include RandAugment [8], mixup [57], cutmix [55], label smoothing [44], layer scale [47], random erasing [58], exponential moving average (EMA) [39], and stochastic depth [17]. These techniques are inspired by the *timm* library [50] and methodologies from Touvron et al. [45]. Detailed descriptions of these hyperparameters are provided in Appendix C.

**Model description.** Mirroring our approach with the CIFAR dataset, we integrate quadratic neurons into a single layer of the ConvNeXt model [34] to conserve computational resources. This adaptation results in the creation of three distinct models: Dit-ConvNeXt-T, Dit-ConvNeXt-S, and Dit-ConvNeXt-B. Specific details about the layer replacement are discussed in the ablation study.

**Results.** Table 3 presents a comparative analysis of Dit-ConvNeXts against state-of-the-art models from the existing literature. Compared to the original ConvNeXt counterparts, our Dit-ConvNeXts exhibit an average increase of 0.5% in top-1 accuracy, with only a marginal average increase of 1% rise

Table 3: Dit-ConvNeXts versus state-of-the-art (SOTA) models on ImageNet-1K. All models listed in the table are trained and validated at a resolution of $224 \times 224$.

| Arch. | Model | # Param. | FLOPs | Top-1 acc. (%) |
|---|---|---|---|---|
| Transformers | Swin-T [33] | 29M | 4.5G | 81.3 |
| | DeiT-S [46] | 22M | 4.6G | 79.8 |
| State Space Models | VMamba-T [32] | 22M | 5.6G | 82.2 |
| | VideoMamba-S [27] | 26M | 4.3G | 81.2 |
| CNNs | ResNet-50 [51] | 26M | 4.1G | 80.4 |
| | SLaK-T [31] | 30M | 5.0G | 82.5 |
| | QuadraNet36-T [53] | 24M | 4.1G | 82.2 |
| | DeepMAD-29M [42] | 29M | 4.5G | 82.5 |
| | ConvNeXt-T [34] | 29M | 4.5G | 82.1 |
| | Dit-ConvNeXt-T | 29M | 5.0G | **82.6** |
| Transformers | Swin-S [33] | 50M | 8.7G | 83.0 |
| State Space Models | VMamba-S [32] | 44M | 11.2G | 83.5 |
| | VideoMamba-M [27] | 74M | 12.7G | 82.8 |
| CNNs | ResNet-101 [51] | 45M | 7.8G | 81.5 |
| | ResNet-152 [51] | 60M | 11.5G | 82.0 |
| | SLaK-S [31] | 55M | 9.8G | 83.8 |
| | QuadraNet36-S [53] | 50M | 8.9G | 83.8 |
| | DeepMAD-50M [42] | 50M | 8.7G | **83.9** |
| | ConvNeXt-S [34] | 50M | 8.7G | 83.1 |
| | Dit-ConvNeXt-S | 50M | 9.2G | 83.6 |
| Transformers | Swin-B [33] | 88M | 15.4G | 83.5 |
| | DeiT-B [46] | 87M | 17.6G | 81.8 |
| State Space Models | VMamba-B [32] | 75M | 18.0G | 83.7 |
| CNNs | SLaK-B [31] | 95M | 17.1G | 84.0 |
| | QuadraNet36-B [53] | 90M | 15.8G | 84.1 |
| | DeepMAD-89M [42] | 89M | 15.4G | 84.0 |
| | ConvNeXt-B [34] | 89M | 15.4G | 83.8 |
| | Dit-ConvNeXt-B | 90M | 16.7G | **84.2** |

in parameter count. Moreover, our models maintain competitive performance against Transformer-based, SSM-based, and CNN-based architectures, highlighting the efficacy and adaptability of Dit-ConvNeXts. Figure 4 further shows that our model consistently improves in accuracy as the model size increases, while other methods tend to saturate, indicating superior scaling property for our model.

## 4.3 Ablation study

### 4.3.1 Dit-CNNs capture data correlation

The expectation of the quadratic term for Gaussian variables can be expressed as follows:

$$E_{x \sim N(\mu, \Sigma)} \left[ x^T A x \right] = \mu^T A \mu + \mathrm{tr}(A\Sigma).$$

This equation highlights that the term $\mathrm{tr}(A\Sigma)$ encapsulates the information derived from data correlation. Consequently, Table 4 examines the impact of this term on the performance of Dit-CNNs in tackling complex tasks. The data presented in Table 4 show a notable decline in accuracy for Dit-CNNs when this term is omitted. This underscores the significance of the quadratic neurons within Dit-CNNs, demonstrating their pivotal role in effectively capturing data correlation, which is essential for task performance.

Table 4: The performance of Dit-CNNs and their counterparts, from which the covariance term $\text{tr}(A\Sigma)$ and the quadratic term $x^T A x$ are omitted in quadratic neurons ($\Sigma$ is estimated from training samples).

| Model | Dataset | Performance (Original) | Performance (minus $\text{tr}(A\Sigma)$) | Performance (minus $x^T A x$) |
|---|---|---|---|---|
| Dit-ResNet-32 | CIFAR-10 | 93.17% | 37.78% | 12.11% |
| Dit-ResNet-32 | CIFAR-100 | 69.68% | 22.33% | 1.32% |
| Dit-ConvNeXt-T | ImageNet-1K | 82.6% | 73.7% | 70.6% |

### 4.3.2 Incorporate quadratic neurons with minimal computational overhead

To manage computational costs effectively, only three layers within the ConvNeXt architecture have been identified as suitable candidates for the integration of quadratic neurons, as depicted in Figure 5. We explore the most effective layer for the integration of quadratic neurons and elucidate the rationale behind this choice. Figure 5 indicates that substituting traditional neurons with quadratic neurons in the first layer of the block 3 yields the most significant performance improvement. Additionally, a negative correlation is observed between the model's final accuracy and the accuracy after omitting the quadratic term from Dit-ConvNeXt-T post-training. This correlation underscores the critical influence of quadratic neurons on the model's efficacy. These findings suggest that Dit-CNNs demonstrating enhanced performance are those where quadratic neurons play a crucial role, highlighting the superior generalization capabilities of quadratic neurons compared to traditional neurons.

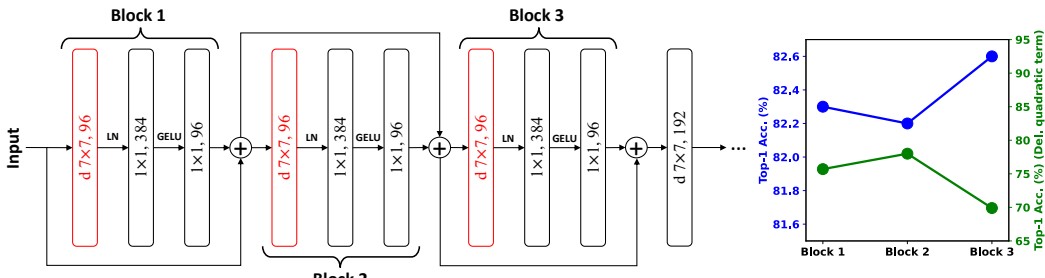

Figure 5: **Left:** Structure of ConvNeXt highlighting three candidate layers (in red) for integrating quadratic neurons. **Right:** ImageNet-1k performance on different Dit-ConvNeXt-T, blue dots indicates top-1 accuracy (left blue vertical axis) while green dots indicates the accuracy post-removal of the quadratic term from Dit-ConvNeXt-T after training (right green vertical axis).

### 4.3.3 Channel/Pixel-wise quadratic neuron utilizations on ImageNet-1K

Table 5: Comparison of quadratic neurons between channel-wise and pixel-wise

| Model | Channel/Pixel-wise | Top-1 acc. (%) |
|---|---|---|
| ConvNeXt-T | Channel-wise (Our Method) | **82.6** |
| | Pixel-wise | 82.2 |
| ConvNeXt-S | Channel-wise (Our Method) | **83.6** |
| | Pixel-wise | 82.9 |
| ConvNeXt-B | Channel-wise (Our Method) | **84.2** |
| | Pixel-wise | 83.7 |

As previously discussed, our Dit-CNNs employ quadratic neurons in a channel-wise manner, which provides a clear biological interpretation, as detailed in Section 4. Meanwhile, employing quadratic neurons on a pixel-wise basis, akin to the quadratic convolution outlined in [20], [35], and [60], leads to a scenario where dendritic integration occurs only when neurons receive synaptic inputs from the

same neuron. This setup simplifies interactions between different neurons to a linear approximation, a method that diverges from biological plausibility within the brain. The superior performance achieved through the channel-wise application of quadratic neurons, as evidenced in Table 5, underscores the efficacy of our model. This result supports the hypothesis that models which more closely mirror brain-like mechanisms tend to exhibit enhanced performance.

## 5 Conclusion

In this paper, we first provide a theoretical demonstration of the computational advantages of quadratic neurons in capturing internal correlation within structured data. These neurons model dendritic integration rules observed in biological neurons. Our empirical evaluations using the MNIST and Arabic MNIST datasets for few-shot learning validate our theoretical assertions. Drawing from the biological interpretation of CNNs, we introduce a biologically plausible method to integrate quadratic neurons into CNN architectures, resulting in Dit-CNNs. These Dit-CNNs not only exhibit significant performance enhancements with minimal modifications to their original counterparts but also compete favorably with state-of-the-art models. The potential applicability of our approach to other architectures, such as Deep-MAD [42], hints at further performance improvements. The promising results of this research underscore the vast potential of brain-inspired models. Given that the quadratic integration rule of neurons could be confined to specific brain areas [15, 28], future electrophysiological experiments could reveal other neuronal integration rules. Our findings could guide the development of brain-inspired deep neural networks by incorporating various integration rules corresponding to different brain areas in distinct layers. Additionally, how to theoretically analyze these new brain-inspired models will be an important issue. It is our aim to extend our analysis concerning high-order statistical information to explore the generalization error of these innovative brain-inspired models. We hope that our work will stimulate further investigations into brain-inspired models, ultimately contributing to the quest for artificial general intelligence (AGI).

## Limitations and Discussions

**Theoretical results for quadratic neurons.** Our theoretical framework is established on the assumption that training samples are normally distributed, a simplification that might not fully encapsulate the complexity inherent in real-world datasets. Despite this assumption, our models have demonstrated exceptional performance on various tasks including ImageNet-1K. This underscores the potential of quadratic neurons in capturing correlation within more intricate data distributions. Future efforts will focus on developing a more comprehensive theoretical foundation to better understand the mechanisms through which quadratic neurons achieve this capability.

**Computational cost of quadratic neurons.** While our Dendritic integration inspired Convolutional Neural Networks (Dit-CNNs) strategically limit the increase in the number of learnable parameters by selectively incorporating quadratic neurons in a singular layer, this modification undeniably raises the computational complexity, as evidenced by an increase in Floating Point Operations (FLOPs). Nevertheless, given the substantial enhancements our Dit-CNNs contribute to model performance, it warrants further investigation into optimizing the efficiency of quadratic neuron deployment. For instance, drawing inspiration from the inherent sparsity observed in the quadratic coefficients of biological neurons—specifically, the pronounced quadratic interactions among synaptic inputs on the same dendritic branch [29]—it is conceivable to predefine a sparse configuration for the quadratic coefficients within our Dit-CNNs. This approach could potentially reduce computational demands while maintaining performance gains.

## Acknowledgments

We thank Ruoyu Sun for his helpful discussions and Zhi-Qin John Xu for his comments. This work was supported by Science and Technology Innovation 2030 - Brain Science and Brain-Inspired Intelligence Project with Grant No. 2021ZD0200204; National Natural Science Foundation of China Grant 12271361, 12250710674 (S.L.); National Natural Science Foundation of China with Grant No. 12225109, 12071287 (D.Z.) and the Student Innovation Center at Shanghai Jiao Tong University (S.L., D.Z.).

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

# A   Proofs of theorems for binary classification task

Assume that the data points of two classes are equally sampled from two distinct non-degenerate normal distributions: $class_1 \sim N(\mu_1, \Sigma_1)$, $class_2 \sim N(\mu_2, \Sigma_2)$, where $\mu_j \in \mathbb{R}^d$, $\Sigma_j \in \mathbb{R}^{d \times d}$ ($j = 1, 2$). The optimal classifier $y_{opt}(x) : \mathbb{R}^d \to \{1, 2\}$ is then defined based on the probability of sampling any point $x$ from these distributions. Specifically, the classifier can be expressed as:

$$y_{opt}(x) = \arg\max_{j \in \{1,2\}} p_j(x), \tag{5}$$

where $p_j(x)$ denotes the probability (density function) of sampling point $x$ from the distribution $N(\mu_j, \Sigma_j)$. The decision rule effectively assigns $x$ to the class with the highest probability density, reflecting the most likely class membership based on the normal distribution parameters.

If a single quadratic neuron, as described in Equation (1) ($f(x) = x^T A x + w \cdot x + b$, where $A$ is a symmetric matrix), is employed to solve the binary classification task described above, we can prove the following theorems.

## A.1   Existence of the optimal solution

**Theorem 3.** *(Existence) The critical points with respect to the cross-entropy loss $L(A, w, b)$ are given as follows:*

$$A^* = \Sigma_1^{-1} - \Sigma_2^{-1}, \; w^* = -2(\Sigma_1^{-1}\mu_1 - \Sigma_2^{-1}\mu_2), \; b^* = \mu_1^T \Sigma_1^{-1}\mu_1 - \mu_2^T \Sigma_2^{-1}\mu_2 + \log\left(\frac{|\Sigma_1|}{|\Sigma_2|}\right).$$

*i.e.*

$$\frac{\partial L}{\partial A}\Big|_{A^*, w^*, b^*} = 0, \quad \frac{\partial L}{\partial w}\Big|_{A^*, w^*, b^*} = 0, \quad \frac{\partial L}{\partial b}\Big|_{A^*, w^*, b^*} = 0,$$

*where*

$$L(A, w, b) = \frac{1}{2}\left[E_{x \sim class_1}\left(\log(1 + e^{f(x)})\right) + E_{x \sim class_2}\left(\log(1 + e^{-f(x)})\right)\right].$$

*Moreover, the corresponding classifier generated by this formula is the same as the theoretically optimal classifier in Equation* (5).

*Proof.* We express the generalized cross-entropy loss $L(A, w, b)$ as:

$$L(A, w, b) = \frac{1}{2}\int_{\mathbb{R}^d}\left(\log(1 + e^{f(x)})p_1(x) + \log(1 + e^{-f(x)})p_2(x)\right) d\mu.$$

The gradients are computed explicitly as follows:

$$\frac{\partial L}{\partial A} = \frac{1}{2}\int_{\mathbb{R}^d}\left(\frac{e^{f(x)}}{1 + e^{f(x)}}p_1(x) - \frac{1}{1 + e^{f(x)}}p_2(x)\right)\frac{\partial f}{\partial A} d\mu,$$

and similarly for $\frac{\partial L}{\partial w}$ and $\frac{\partial L}{\partial b}$. Given the probability density function of the multivariate normal distribution:

$$p_j(x) = (2\pi)^{-d/2}|\Sigma_j|^{-1/2}\exp\left(-\frac{1}{2}(x - \mu_j)^T\Sigma_j^{-1}(x - \mu_j)\right) \quad j \in \{1, 2\}.$$

If the parameters $A, w, b$ are set as:

$$A = \Sigma_1^{-1} - \Sigma_2^{-1}, \; w = -2(\Sigma_1^{-1}\mu_1 - \Sigma_2^{-1}\mu_2), \; b = \mu_1^T \Sigma_1^{-1}\mu_1 - \mu_2^T \Sigma_2^{-1}\mu_2 + \log\left(\frac{|\Sigma_1|}{|\Sigma_2|}\right),$$

one can obtain

$$f(x) = x^T(\Sigma_1^{-1} - \Sigma_2^{-1})x - 2(\mu_1^T\Sigma_1^{-1} - \mu_2^T\Sigma_2^{-1})x + \mu_1^T\Sigma_1^{-1}\mu_1 - \mu_2^T\Sigma_2^{-1}\mu_2 + \log\left(\frac{|\Sigma_1|}{|\Sigma_2|}\right)$$

which ensures that

$$e^{f(x)} = \frac{p_2(x)}{p_1(x)}, \; \forall x \in \mathbb{R}^d.$$

Consequently,

$$\frac{e^{f(x)}}{1+e^{f(x)}}p_1(x) - \frac{1}{1+e^{f(x)}}p_2(x) = 0, \ \forall x \in \mathbb{R}^d \Rightarrow \frac{\partial L}{\partial A} = 0, \frac{\partial L}{\partial w} = 0, \frac{\partial L}{\partial b} = 0.$$

The classifier generated by the quadratic neuron is defined as:

$$y_{model}(x) = \begin{cases} 1 & \text{if } f(x) < 0, \\ 2 & \text{if } f(x) > 0. \end{cases}$$

Therefore,

$$y_{model}(x) = 1 \Leftrightarrow y_{opt}(x) = \underset{j \in \{1,2\}}{\arg\max}\, p_j(x) = 1.$$

Similarly, $y_{model}(x) = 2$ implies $y_{opt}(x) = 2$. Hence, $y_{model}(x) = y_{opt}(x)$ for all $x$ in $\mathbb{R}^d$, demonstrating the consistency between these two classifiers. $\qquad\square$

## A.2   Uniqueness of the optimal solution

To prove the theorem for uniqueness, we first introduce the following preliminary lemmas:

**Lemma 1.** *Let $\mathcal{M}(\mathbb{R}^d)$ denote the Lebesgue $\sigma$-algebra on $\mathbb{R}^d$. Consider $f(x)$, a Lebesgue measurable function on $\mathbb{R}^d$, satisfying:*

$$\int_\Omega f(x)\,d\mu = 0, \ \forall \Omega \in \mathcal{M}(\mathbb{R}^d).$$

*It follows that $f(x) = 0$ almost everywhere on $\mathbb{R}^d$.*

*Proof.* To demonstrate this via contradiction, let us assume, without loss of generality, that:

$$\mu(\{x \mid f(x) > 0\}) > 0.$$

Since

$$0 < \mu(\{x \mid f(x) > 0\}) = \mu(\bigcup_{n=1}^{+\infty}\{x \mid f(x) > \frac{1}{n}\}) \le \sum_{n=1}^{+\infty}\mu(\{x \mid f(x) > \frac{1}{n}\}),$$

then we know there exist $n_0 \in \mathbb{Z}^*$ such that $\mu(\{x \mid f(x) > \frac{1}{n_0}\}) > 0$. If we set $\Omega_0 = \{x \mid f(x) > \frac{1}{n_0}\} \in \mathcal{M}(\mathbb{R}^d)$, then by the property of $f(x)$ we can derive:

$$0 = \int_{\Omega_0} f(x)\,d\mu > \frac{\mu(\{x \mid f(x) > \frac{1}{n_0}\})}{n_0} > 0.$$

Hence, we have reached a contradiction. $\qquad\square$

**Lemma 2.** *Assume $f(x) = x^T A x + w \cdot x + b$, a Lebesgue measurable function, equals zero almost everywhere ($f(x) = 0$ a.e.), with $A$ being a symmetric matrix. It then follows that $A = 0$, $w = 0$, and $b = 0$.*

*Proof.* Since $f(x)$ is continuous, we know $f(x) = 0$ for $\forall x \in \mathbb{R}^d$. Then we have:

$$b = f(0) = 0.$$

Setting $x = \varepsilon w$ for some $\varepsilon > 0$, we obtain:

$$\|w\|_2^2 + \varepsilon w^T A w = 0,$$

letting $\varepsilon$ go to zero, we have:

$$\|w\|_2^2 = 0 \Rightarrow w = 0.$$

Having established that $f(x) = x^T A x$, consider $x = e_i$, where $e_i$ is the unit vector with a value of 1 at the $i$-th position and 0s elsewhere. Evaluating the function at this vector, we have $f(e_i) = e_i^T A e_i = a_{ii} = 0$. Subsequently, by letting $x = e_i + e_j$, where $e_j$ is another unit vector corresponding to the $j$-th position, we obtain the following equation:

$$a_{ii} + a_{ij} + a_{ji} + a_{jj} = 0.$$

Since $a_{ii} = a_{jj} = 0$ and $a_{ij} = a_{ji}$ ($A$ is a symmetric matrix), we know:

$$a_{ij} = 0, \ \forall i, j \in \{1, 2, \ldots, d\} \Rightarrow A = 0.$$

$\qquad\square$

**Theorem 4.** *(Uniqueness) Consider the conditional cross-entropy loss defined on a set $\Omega \in \mathcal{M}(\mathbb{R}^d)$, where $\mathcal{M}(\mathbb{R}^d)$ denotes the Lebesgue $\sigma$-algebra on $\mathbb{R}^d$:*

$$L(A, w, b \mid \Omega) = \frac{1}{2} \left[ E_{x \sim class_1, x \in \Omega} \left( \log(1 + e^{f(x)}) \right) + E_{x \sim class_2, x \in \Omega} \left( \log(1 + e^{-f(x)}) \right) \right],$$

*with*

$$E_{x \sim class_j, x \in \Omega} \left( g(x) \right) = \int_\Omega g(x) p_j(x) \, d\mu,$$

*then the unique solution satisfies*

$$\frac{\partial L(A, w, b \mid \Omega)}{\partial A} \Big|_{A^*, w^*, b^*} = 0, \ \frac{\partial L(A, w, b \mid \Omega)}{\partial w} \Big|_{A^*, w^*, b^*} = 0, \ \frac{\partial L(A, w, b \mid \Omega)}{\partial b} \Big|_{A^*, w^*, b^*} = 0$$

*for every $\Omega \in \mathcal{M}(\mathbb{R}^d)$ given by:*

$$A^* = \Sigma_1^{-1} - \Sigma_2^{-1}, \ w^* = -2(\Sigma_1^{-1}\mu_1 - \Sigma_2^{-1}\mu_2), \ b^* = \mu_1^T \Sigma_1^{-1} \mu_1 - \mu_2^T \Sigma_2^{-1} \mu_2 + \log\left( \frac{|\Sigma_1|}{|\Sigma_2|} \right).$$

*Proof.* Following the derivation outlined in the proof of Theorem 3, we obtain:

$$\frac{\partial L(A, w, b \mid \Omega)}{\partial A} = \frac{1}{2} \int_\Omega \left( \frac{e^{f(x)}}{1 + e^{f(x)}} p_1(x) - \frac{1}{1 + e^{f(x)}} p_2(x) \right) x \cdot x^T \, d\mu,$$

$$\frac{\partial L(A, w, b \mid \Omega)}{\partial w} = \frac{1}{2} \int_\Omega \left( \frac{e^{f(x)}}{1 + e^{f(x)}} p_1(x) - \frac{1}{1 + e^{f(x)}} p_2(x) \right) x \, d\mu,$$

$$\frac{\partial L(A, w, b \mid \Omega)}{\partial b} = \frac{1}{2} \int_\Omega \left( \frac{e^{f(x)}}{1 + e^{f(x)}} p_1(x) - \frac{1}{1 + e^{f(x)}} p_2(x) \right) d\mu.$$

Hence, if $A^*, w^*$, and $b^*$ satisfy:

$$\frac{\partial L(A, w, b \mid \Omega)}{\partial A} \Big|_{A^*, w^*, b^*} = 0, \ \frac{\partial L(A, w, b \mid \Omega)}{\partial w} \Big|_{A^*, w^*, b^*} = 0, \ \frac{\partial L(A, w, b \mid \Omega)}{\partial b} \Big|_{A^*, w^*, b^*} = 0$$

for every $\Omega \in \mathcal{M}(\mathbb{R}^d)$. Subsequently, invoking Lemma 1 yields:

$$e^{x^T A^* x + (w^*)^T x + b^*} = \frac{p_2(x)}{p_1(x)}, \ \ a.e.$$

Simplifying this equation, we have

$$x^T(\Sigma_1^{-1} - \Sigma_2^{-1} - A^*)x - \left[ 2(\mu_1^T \Sigma_1^{-1} - \mu_2^T \Sigma_2^{-1}) + (w^*)^T \right] x + \mu_1^T \Sigma_1^{-1} \mu_1 - \mu_2^T \Sigma_2^{-1} \mu_2 + \log\left( \frac{|\Sigma_1|}{|\Sigma_2|} \right) - b^* = 0,$$

which holds almost everywhere. Applying Lemma 2, we obtain:

$$A^* = \Sigma_1^{-1} - \Sigma_2^{-1}, \ w^* = -2(\Sigma_1^{-1}\mu_1 - \Sigma_2^{-1}\mu_2), \ b^* = \mu_1^T \Sigma_1^{-1} \mu_1 - \mu_2^T \Sigma_2^{-1} \mu_2 + \log\left( \frac{|\Sigma_1|}{|\Sigma_2|} \right).$$

$\square$

## B   Quadratic neurons capture correlation on multi-class classification task

For a classification task with $k$ classes, assume the training samples for these classes are equally drawn from $k$ normal distributions: $class_j \sim N(\mu_j, \Sigma_j), j \in [k]$, the optimal classifier $y_{opt}(x) : \mathbb{R}^d \to \{1, 2, \ldots, k\}$ is defined by:

$$y_{opt}(x) = \arg\max_{j \in [k]} p_j(x), \tag{6}$$

where $p_j(x)$ represents the probability (density function) of sampling point $x$ from the normal distribution $N(\mu_j, \Sigma_j)$. Employing $k$ quadratic neurons for this multi-class classification task allows for the derivation of a theorem analogous to the one discussed in the previous section:

**Theorem 5.** *(Existence) The critical points with respect to the cross-entropy loss $L(\theta)$ are given as follows:*

$$A_j^* = \Sigma_j^{-1}, \ w_j^* = -2\Sigma_j^{-1}\mu_j, \ b^* = \mu_j^T \Sigma_j^{-1} \mu_j + \log(|\Sigma_j|), \ j \in [k],$$

*where*

$$L(\theta) = \frac{1}{k} \sum_{j=1}^{k} E_{x \sim class_j} \left[ \log \left( 1 + \sum_{i=1, i \neq j}^{k} e^{f_i(x) - f_j(x)} \right) \right].$$

*Moreover, the corresponding classier generated by this formula is the same as the theoretically optimal classifier as defined in Equation* (6).

*Proof.* The gradients can be computed explicitly as follows:

$$\frac{\partial L}{\partial A_j} = \frac{1}{k} \int_{\mathbb{R}^d} \left( \sum_{i=1, i \neq j}^{k} \frac{e^{f_j(x)}}{\sum_{i=1}^{k} e^{f_i(x)}} p_i(x) - \sum_{i=1, i \neq j}^{k} \frac{e^{f_i(x)}}{\sum_{i=1}^{k} e^{f_i(x)}} p_j(x) \right) x \cdot x^T \, d\mu, \ j \in [k],$$

and similarly for $\frac{\partial L}{\partial w_j}$ and $\frac{\partial L}{\partial b_j}$ that can be calculated. From the derivation outlined in the proof of Theorem 3 and the choice for critical points as follows

$$A_j^* = \Sigma_j^{-1}, \ w_j^* = -2\Sigma_j^{-1}\mu_j, \ b^* = \mu_j^T \Sigma_j^{-1} \mu_j + \log(|\Sigma_j|), \ j \in [k],$$

then we have

$$e^{f_j(x) - f_i(x)} = \frac{p_j(x)}{p_i(x)}, \ \forall x \in \mathbb{R}^d, \ \forall i, j \in [k],$$

which is equivalent to

$$\frac{e^{f_j(x)}}{\sum_{i=1}^{k} e^{f_i(x)}} p_i(x) = \frac{e^{f_i(x)}}{\sum_{i=1}^{k} e^{f_i(x)}} p_j(x), \ \forall x \in \mathbb{R}^d, \ \forall i, j \in [k].$$

Thus, we can derive:

$$\frac{\partial L}{\partial A_j} = 0, \ \frac{\partial L}{\partial w_j} = 0, \ \frac{\partial L}{\partial b_j} = 0, \ \forall j \in [k].$$

The classifier generated by quadratic neurons is defined as:

$$y_{model}(x) = \arg\max_{j \in [k]} f_j(x),$$

Therefore,

$$y_{model}(x) = j \Leftrightarrow f_j(x) > f_i(x), \ \forall i \in [k]/\{j\} \Leftrightarrow y_{opt}(x) = j.$$

Hence, $y_{model}(x) = y_{opt}(x)$ for all $x$ in $\mathbb{R}^d$, demonstrating the consistency between these two classifiers. $\square$

Theorem 5 demonstrates that in multi-class classification tasks, each neuron captures data correlation by directly relating to the covariance matrix of their respective class. This concept is empirically validated through numerical experiments on the MNIST dataset, as depicted in Figure 6. The eigenvectors associated with the largest eigenvalues of the covariance matrices, representing the principal components of each class's distribution, reveal the spatial correlation within that class's data. The marked similarity between these eigenvectors and the eigenvectors of the quadratic integration matrices of neurons confirms that quadratic neurons effectively capture correlation from the data distribution. This intrinsic ability of quadratic neurons to capture correlations provides a possible explanation for the superior generalization capability of quadratic neurons.

## C Experimental settings of ImageNet training

The training settings for our Dit-ConvNeXt models on ImageNet-1K are detailed in Table 6. While all Dit-ConvNeXt variants adhere to a unified configuration, certain parameters—namely, the stochastic depth rate, learning rate, and weight decay—are individually customized for each model variant. These hyperparameters were optimized based on performance outcomes, utilizing a grid search over potential learning rates from the set $\{0.004, 0.006, 0.008\}$ and weight decay values options within $\{0.04, 0.06, 0.08\}$.

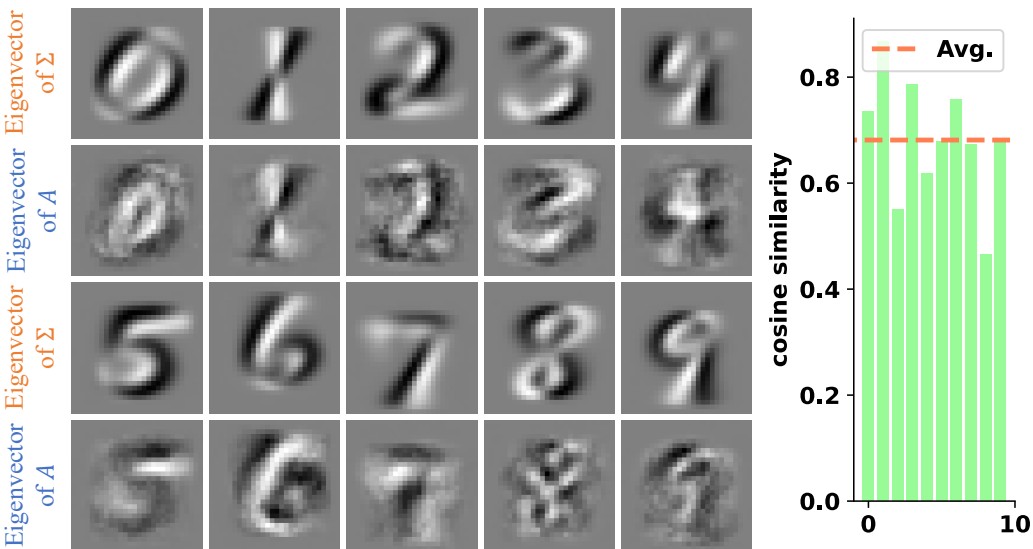

Figure 6: Comparison of Eigenvectors between covariance matrices $\Sigma_j$ and quadratic weights $A_j$ for $j \in \{0, 1, 2, 3, 4, 5, 6, 7, 8, 9\}$. **Left**: Visualization of eigenvectors corresponding to the largest eigenvalue of $\Sigma_j$ alongside the most similar eigenvectors of $A_j$. **Right**: Cosine similarity metrics for ten eigenvector pairs depicted on the left.

Table 6: ImageNet-1K training settings.

| Training config | Dit-ConvNeXt-T | Dit-ConvNeXt-S | Dit-ConvNeXt-B |
|---|---|---|---|
| Optimizer | AdamW | AdamW | AdamW |
| Base learning rate | 6e-3 | 6e-3 | 4e-3 |
| Weight decay | 0.08 | 0.06 | 0.06 |
| Optimizer momentum | $\beta_1, \beta_2 = 0.9, 0.999$ | $\beta_1, \beta_2 = 0.9, 0.999$ | $\beta_1, \beta_2 = 0.9, 0.999$ |
| Batch size | 4096 | 4096 | 4096 |
| Training epochs | 300 | 300 | 300 |
| Learning rate schedule | cosine | cosine | cosine |
| Warmup epochs | 20 | 20 | 20 |
| Warmup schedule | linear | linear | linear |
| RandAugment [8] | rand-m9-mstd0.5 | rand-m9-mstd0.5 | rand-m9-mstd0.5 |
| Mixup [57] | 0.8 | 0.8 | 0.8 |
| Cutmix [55] | 1.0 | 1.0 | 1.0 |
| Random erasing [58] | 0.25 | 0.25 | 0.25 |
| Label smoothing [44] | 0.1 | 0.1 | 0.1 |
| Stochastic depth [17] | 0.1 | 0.4 | 0.4 |
| Layer scale [47] | 1e-6 | 1e-6 | 1e-6 |
| Exp. mov. avg. (EMA) [39] | 0.9998 | 0.9998 | 0.9998 |

