# OpenReview forum: "Dendritic Integration Inspired Artificial Neural Networks Capture Data Correlation"
_NeurIPS.cc/2024/Conference — NeurIPS 2024 poster_

### Official Review · Reviewer_gq5j · 2024-07-03

**Soundness:** 3
**Presentation:** 3
**Contribution:** 3
**Rating:** 6
**Confidence:** 4

**Summary:**

This study explores incorporating channel-wise quadratic neuron, as inspired by dendritic nonlinearity, into artifical neural networks to improve model performance. These models show competitive performance on datasets like CIFAR, ImageNet-1K while maintaining simplicity and efficiency. The theoretical and experimental results highlight the potential of quadratic neurons in enhancing model performance.

**Strengths:**

The paper presents a sound and well-supported exploration of quadratic neurons in ANNs. The theoretical analysis gives a good intuitive demonstration on the reason behind the benefit of quadratic neurons. The experimental verification is clear and show clear advantage given by their architecture.
Overall the paper is well-written and clearly explained.

**Weaknesses:**

The theortical explaination is limited to highly simplified setting.
It is not clear if the emperical analysis on imagenet part is a fair one, give quite sophicated data augumentation is used for their models.

**Questions:**

See above.

**Limitations:**

Yes.

---

> ### Author Rebuttal · Authors · 2024-08-06
>
> ## Weaknesses
> - "The theortical explaination is limited to highly simplified setting. "
>
> Thank you for your comment. Our theoretical analysis aims to emphasize that quadratic neurons inherently capture second-order information from training samples, which enhances their generalization capabilities compared to conventional neurons. In more complex scenarios, such as CIFAR-10 and ImageNet using deep CNNs, deriving theoretical results becomes challenging. However, theoretical analysis under simplified setting provided insights into why quadratic neurons perform better. Therefore, we conduct ablation study in Section 4.3.1 to show that quadratic neurons indeed capture second-order information for classification when they performs better in these cases, further supporting our conclusions.
>
> - "It is not clear if the emperical analysis on imagenet part is a fair one, give quite sophicated data augumentation is used for their models."
>
> Thank you for your valuable feedback. The comparison is fair from two perspectives. Firstly, Our Dit-ConvNeXt employed the same data augmentation techniques as the original ConvNeXt, which indicates that our model does not require significant modifications to achieve notable improvements. Secondly, the state-of-the-art (SOTA) models we compared in Table 3 of the main text also utilize extensive data augmentations and tuning to achieve their results.

---

### Official Review · Reviewer_mc29 · 2024-07-08

**Soundness:** 3
**Presentation:** 2
**Contribution:** 3
**Rating:** 7
**Confidence:** 1

**Summary:**

The paper introduces a new biologically inspired neural network architecture. Rather than using linear layers followed by a nonlinear activation function, the authors propose a quadratic model instead in the inputs. This form is said to explicitly model the covariance between input features offering better accuracy and generalization.

**Strengths:**

The authors present improved accuracy with fewer parameters than existing state of the art models by simply replacing a few layers.
The authors have conducted a thorough series of experiments

**Weaknesses:**

The relation to biological plausibility is unclear and the link is tenuous.

**Questions:**

Have the authors look into the structured learned by the quadratic term?
How do the CNN filters differ from those previously learned?
Did the authors compare to other smoother activation functions such as gelu?

**Limitations:**

The limitations are not outlined in the main text but are in the appendix instead.
It would be beneficial to include these in the main text.

---

> ### Author Rebuttal · Authors · 2024-08-06
>
> ## Weaknesses
> Thank you for the valuable comments, we have provided a biological interpretation of our Dit-CNN in global rebuttal.
>
> ## Questions
> - "Have the authors look into the structured learned by the quadratic term?"
>
> Thank you for your valuable question. Following your suggestion, we have examined the distribution of the trained quadratic coefficients, as shown in Figure 3.B of the rebuttal materials. We find that most of the coefficients are close to zero, suggesting the potential to further reduce the computational cost of our model. Additionally, the distribution is symmetric and confined to a reasonable range, without extremely large or small values.
>
> - "How do the CNN filters differ from those previously learned?"
>
> Thank you for your question. Following your suggestion, we visualize the CNN filters of the original ConvNeXt and our proposed Dit-ConvNeXt, as shown in Figure 2.B of the rebuttal materials. This visualization highlights the differences in training results between the two models, indicating that our Dit-CNN is not merely a small modification of the original CNNs.
>
> - "Did the authors compare to other smoother activation functions such as gelu?"
>
> Thank you for your helpful question. Following your suggestion, we compared our quadratic neuron with the smoother activation function GeLU on binary classification and the MNIST dataset, as shown in Figure 3.A of the rebuttal materials. Our results demonstrate that while GeLU activation performs better than ReLU in the binary classification task, it still does not achieve the optimal results seen with quadratic neurons. Additionally, GeLU activation performs even worse on the MNIST dataset. For more complex datasets, ConvNeXt already utilizes GeLU activation, and the results for this case are presented in Table 3 of the main text. These results are consistent with our theoretical analysis: the computational advantage of quadratic neurons arises from their direct second-order interactions between inputs, enabling them to better capture data correlations, rather than from the smoothness of the quadratic function.
>
> ## Limitations
> Thank you for your advice. We will move the limitations to the main text in a later revision.

---

> ### Comment · Reviewer_mc29 · 2024-08-13
>
> I have read the rebuttals. I have updated my score.

---

> > ### Author Response · Authors · 2024-08-13
> >
> > We are delighted to see that our clarification and rebuttal were well-received, and we appreciate the increase in the score. Thank you once again for your careful review and valuable recommendations.

---

### Official Review · Reviewer_8aT5 · 2024-07-10

**Soundness:** 3
**Presentation:** 3
**Contribution:** 3
**Rating:** 7
**Confidence:** 5

**Summary:**

This is an interesting paper looking at quadratic neurons and how they impact performance and/or learning rate of ANNs. The quadratic integration is loosely linked to dendritic integration properties of pyramidal neurons in cortex (though it is unclear whether any real resemblence should be granted). This increase in the degree of nonlinearity , when implemented in CNNS (called Dit-CNNs) works to the advantage on benchmarks like imagenet while also retaining the simplicity of conventional CNNs. Of interest and importance are the elegant anaytical solutions exemplified in figure 1 as well as the ablation study shown in table 4. Overall this is an interesting and timely study.

**Strengths:**

This is an interesting study looking at the impact of relatively recent findings in neurobiology . Even if how these neurobiological findings are loosely connected to the quadratic model, how the model and the associated networks are characterized is thorough and insightful. The ablation part as well as the work on how to integrate the quadratic neurons with minimal overhead are nice contributions that further substantiate the author's decision to account for higher terms that could potentially be linked to dendritic integration.

**Weaknesses:**

The authors use the term "dendritic integration" loosely without exploring various ways that such integration can occur (sub- vs. supra-linear). It would be interesting to consider such cases.

**Questions:**

could a potentially more nuanced or bio-plausible implementation of dendritic integration further enhance performance? How would that affect the quadratic integration?

**Limitations:**

it is left somewhat unclear how the dendritic integration along different compartments of a cell and its result in the neuron's output is reflected in quadratic neurons. It would be interesting to look at sub- vs. supra-linear integration effects along the same lines.

---

> ### Author Rebuttal · Authors · 2024-08-06
>
> ## Weaknesses
> Thank you for your suggestion, it is indeed an interesting idea. Previous work has shown that the dendritic bilinear integration rule can account for both sub-linear and supra-linear cases, depending on the sign of the quadratic coefficient. In our approach, we directly incorporate this rule into artificial neural networks (ANNs) without restricting the integration to only sub-linear or supra-linear cases, specifically, we do not fix the sign of the quadratic coefficients during training, allowing both sub-linear and supra-linear to happen. An additional observation is that, after training, the quadratic coefficients exhibit both positive and negative components, as illustrated in Figure 3.B of the rebuttal materials. This indicates that our models capture both sub-linear and supra-linear effects.
>
> ## Questions
> Thank you for your insightful question. The dendritic bilinear integration rule is primarily restricted to subthreshold regime before the neuron generate spikes. However, there are other properties of dendritic integration, such as dendritic spikes [1], that warrant further exploration. Due to time constraints, we have not fully investigated these properties. However, examining how to incorporate these features into artificial neural networks (ANNs) and how to combine them with the bilinear integration rule could potentially enhance performance. This is an area that deserves future research.
>
> [1] Gidon, Albert, et al. "Dendritic action potentials and computation in human layer 2/3 cortical neurons." Science 367.6473 (2020): 83-87.
>
> ## Limitations
> Thank you for your valuable suggestion. The dendritic bilinear integration rule already captures the somatic response when synaptic inputs are integrated from different compartments of a neuron. Therefore, we can model quadratic neurons without explicitly considering sub- and supra-linear integration along these compartments. However, we agree that exploring the integration effects in dendritic compartments could deepen our understanding of neuronal computation, and this area deserves further investigation.

---

### Official Review · Reviewer_opWC · 2024-07-11

**Soundness:** 2
**Presentation:** 2
**Contribution:** 2
**Rating:** 4
**Confidence:** 4

**Summary:**

This paper explores the computational benefits of quadratic neurons, which are inspired by the quadratic integration rules of dendrites. The authors first present the theoretical analysis on binary classification for normal distributions, showing the existence and uniqueness of the solution with a single quadratic neuron. Then, a few-shot learning experiment on MNIST and Arabic MNIST is conducted to show the better performance of quadratic neurons under few-shot training samples. Finally, this paper integrates quadratic neurons into CNNs along the channel dimension, and demonstrates the superior performance of the model on several datasets including CIFAR and ImageNet.

**Strengths:**

1. This paper considers the quadratic integration rule inspired by biological dendrites and successfully applies it to advanced artificial neural networks such as ConvNeXt.

2. Experiments show promising performance on relatively large-scale datasets, e.g., ImageNet.

**Weaknesses:**

1. The presentation and organization of the paper are poor and loose. There are also many informal claims without rigorous justification.

(1.1) The link between the theoretical analysis, the so-called “enhanced generalization capabilities”, the few-shot learning advantage, and the integration into CNNs is poor. I do not see that the theoretical analysis of the existence and uniqueness of the solution of a single quadratic neuron under a simplified setting provides effective insight or explanation for the following practice. There is no analysis of deep networks. There is no explanation for the connection between the practice of CNNs and theories.

(1.2) In the discussion of Theorem 1 and 2, it is said using the gradient descent algorithm. However, there is no analysis for training dynamics or learnability in current Theorems.

(1.3) There is no formal definition or rigorous justification for the so-called “superior generalization capability over traditional neuron”.

(1.4) It is claimed that the integration of quadratic neurons into CNNs along the channel dimension is “biologically plausible”. However, I do not see any detailed justification. Indeed, only considering quadratic integration along the channel dimension does not correspond to synaptic connections between neurons.

2. The novelty and significance of this paper are not clear enough. Quadratic neurons have been studied in many previous works, both from theoretical and practical perspectives, e.g., [1] has shown strong results. While this paper interprets it as inspiration from biological neurons, the implementation does not correspond to the biological computation form. So there should be more differentiation with existing quadratic neuron works. Considering Table 1, the biological interpretation in this paper is not strong enough, and there are also theories in previous works [1]. Only changing how the quadratic operation is used makes little contribution.

3. There is no formal definition and analysis showing “enhance data correlation” in the title.

[1] Fan, F. L., Dong, H. C., Wu, Z., Ruan, L., Zeng, T., Cui, Y., & Liao, J. X. (2023). One neuron saved is one neuron earned: On parametric efficiency of quadratic networks. arXiv preprint arXiv:2303.06316.

**Questions:**

Will incorporating quadratic neurons into multiple layers continually increase the performance? How is the suitable layer candidate identified?

**Limitations:**

The authors discussed limitations in Appendix D.

---

> ### Author Rebuttal · Authors · 2024-08-06
>
> ## Weaknesses
> 1. Thank you for your insightful feedback, and I apologize for any lack of clarity in our presentation.
>
> (1.1) Our main analysis aims to  emphasizes that quadratic neurons inherently capture second-order information from training samples, which enhances their generalization capabilities compared to conventional neurons.
>
> To support this, we provide a theoretical proof demonstrating how quadratic neuron capture second-order information to achieve optimal result in binary classification for two normal distributions characterized by first and second-order moments. And our numerical experiments shows that quadratic neurons indeed effectively capture this information while conventional neurons can not.
>
> We generalize the theorem to multi-class classification tasks (Theorem 5). We also perform numerical experiments on MNIST to verify that quadratic neurons capture second-order information in this case (Appendix B, Figure 5).
>
> In more complex scenarios, such as CIFAR-10 and ImageNet using deep CNNs, deriving theoretical results becomes challenging. However, previous theoretical analyses provided insights into why quadratic neurons perform better. So we conduct ablation study in Section 4.3.1 to show that quadratic neurons indeed capture second-order information for classification when they performs better in these cases, further supporting our conclusions.
>
> (1.2) Thank you for your observation. Theorem 1 elucidates a possible analytical solution (existence of critical point) for gradient flow algorithm. And Theorem 2 establish the uniqueness of that critical points under certain assumptions. Under these conditions, it can be theoretically proven that if the gradient flow algorithm eventually converges, it will drive the parameters to this unique critical point [1][2]. We will clarify this in the updated version. And the numerical results presented in Section 3.1 (Figure 1 with sufficient training samples) further confirm the correspondence between the theoretically identified critical point and the numerically converging one.
>
> [1] Quarteroni, Alfio, Riccardo Sacco, and Fausto Saleri. Numerical mathematics. Vol. 37. Springer Science & Business Media, 2006.
>
> [2] Ambrosio, Luigi, Nicola Gigli, and Giuseppe Savaré. Gradient flows: in metric spaces and in the space of probability measures. Springer Science & Business Media, 2008.
>
> (1.3) The superior generalization capability means the ability of models incorporating quadratic neurons to achieve lower generalization error. For instance, we demonstrate this through the smaller generalization error observed in binary classification tasks where the target function is known, as well as higher test accuracy on the MNIST, CIFAR-10, and ImageNet-1 datasets. We will clarify the meaning of “superior generalization capability over traditional neuron” in the updated version.
>
> (1.4) Thank you for the valuable comments, we have provided a biological interpretation of our Dit-CNN in global rebuttal.
>
> 2. Thank you for your feedback. We clarify the novelty and significance as follows:
> - **Biological Interpretation:** Our channel-wise quadratic model offers a comprehensive biological interpretation, as explained in global rebuttal, whereas other quadratic methods lack such evidence.
> - **Efficiency:** Our channel-wise quadratic form demonstrates significantly higher efficiency, utilizing only about one-third of the number of trainable parameters compared to other quadratic methods. This is detailed in Table 2 of the main text, where we also include a comparison with the model referenced in your review ([12] in the main text).
> - **Scaling Property:** We have added a visualization of the test accuracy comparison in Figure 2.A of the rebuttal materials. This shows that our model consistently improves in accuracy as the model size increases, while other quadratic methods tend to saturate, indicating superior scaling property for our model.
> - **Focus on Generalization:** Our analysis does not concentrate on the universal approximation properties of networks, as seen in the theorems of the paper you provided. Instead, we aim to explain the advantages of quadratic neurons from the generalization capability perspective, emphasizing their ability to capture second-order (correlation) information from data distributions. We believe these insights differentiate our work from existing studies on quadratic neurons and contribute to a various understanding of their practical benefits.
>
> 3. Thank you for your helpful feedback. In this paper, "enhanced data correlation" refers to the model's ability to capture second-order information (correlation) effectively, which is verified through our numerical experiments on MNIST, CIFAR, and ImageNet. To clarify, we will change “enhance data correlation” to “capture data correlation” in the updated version.
>
> ## Questions
> Thank you for your insightful questions.
> - "Will incorporating quadratic neurons into multiple layers continually increase the performance?"
>
> Due to limited time and resources, we have only experimented with incorporating quadratic neurons into multiple layers on CIFAR-10, which resulted in a gradual increase in performance. However, our observations suggested that replacing only one layer is often the best choice for balancing computational cost and performance.
> - "How is the suitable layer candidate identified?"
>
> We discuss this in Section 4.3.2 of the main text. Suitable layer candidates are identified based on the trade-off between computational cost and performance improvement. In our practice, we initially incorporate quadratic neurons into each layer of a fixed architecture, such as ResNet, to evaluate performance in smaller models like ResNet-20. Once a suitable layer for incorporating quadratic neurons is identified, the same replacement is applied to deeper models, such as ResNet-110.

---

> > ### Comment · Reviewer_opWC · 2024-08-13
> >
> > I would like to thank the authors for their detailed responses. I acknowledge that the empirical results (efficiency, scaling property, etc.) are good, which I have listed as strengths, while my concerns remain that the theoretical presentation and claims for biological plausibility are not satisfying. Considering the theoretical part, the authors emphasize that they focus on generalization. However, there is a large gap between "capture second-order information (correlation)" and generalization in general settings, and there is no formal analysis for generalization error. Considering biological plausibility, the quadratic term in Dit-CNN is element-wise spatially rather than considering convolution with receptive field, which means some synapses are used for quadratic integration while some are only summed. This discrepancy is still not explained. I understand that these parts may not be actually necessary for brain-inspired algorithms with good performance, but if the authors claim this as an important part of the contribution, I think the current presentation is not complete and satisfying. So I keep my score for the current version of the paper.

---

> > > ### Author Response · Authors · 2024-08-13
> > >
> > > Thank you very much for your thoughtful feedback and for taking the time to review our rebuttal. We would like to further clarify your concerns.
> > >
> > > Regarding analysis, our theoretical proof demonstrates a clear computational advantage of quadratic neurons: they capture second-order information, which distinguishes them from traditional neurons. Following our observation of a significant improvement in test accuracy in more complex scenarios, we have confirmed that quadratic neurons effectively capture second-order information in these contexts. So the capability of capturing second-order information likely plays an important role in small generalization errors observed in our models.
> > >
> > > Regarding biological plausibility, we had considered the quadratic interactions between convolutions and encountered training challenges, such as gradient explosion. Alternatives such as considering both spatial-wise and channel-wise quadratic interactions, may significantly increase the computational cost (e.g.,  a $3\times 3$ convolution with both spatial-wise and channel-wise quadratic interactions leads to a 81 times increase in FLOPs compared to our Dit-CNN). Therefore, we proposed Dit-CNN, which achieves good performance while maintaining practical computational costs. We believe we are the very first to explore quadratic methods based on the dendritic bilinear integration rule. And Our Dit-CNN is inspired by the visual pathways in the neural system, as illustrated in Figure 1.C of the rebuttal materials (a modification with good performance and low computation cost ).
> > >
> > > We hope these explanations address your concerns. Once again, thank you for your patience and for raising such constructive points.

---

### Official Review · Reviewer_cuyp · 2024-07-24

**Soundness:** 2
**Presentation:** 2
**Contribution:** 3
**Rating:** 5
**Confidence:** 4

**Summary:**

The paper theoretically demonstrate that quadratic neurons inspired from dendritic computing inherently capture correlation within structured data. The quadratic rules are integrated with convolution networks and established the so-called Dit-CNNs. Experiments on CIFAR and ImageNet datasets demonstrates competitive performance of the proposed model.

**Strengths:**

The paper provides theoretically demonstration of critical points existence of parameters of quadratic neuron. An illustrative binary classification experiment is constructed. The quadratic rules are further verified on MLP network with few-shot MNIST learning task and convolution networks in CIFAR and ImageNet classification tasks. These experiments demonstrate competitive performance of the proposed model to other related methods.

**Weaknesses:**

1. The paper demonstrates that there exist critical points for the parameters of quadratic neuron, however, there lacks experimental proof that the training process actually drives these parameters towards the critical points. It would be more convincing if the author could provide such evidence, for datasets with different scales.
2. Compared to previous quadratic dendritic methods, the improvement in accuracy of the proposed model is not so significant, I suggest that the author provide comparisons on computation cost to further demonstrate the advantage of the proposed model.

**Questions:**

1. In the ablation study 4.3.1, table 4, why performance dropping level are significantly different for CIFAR and ImageNet?

**Limitations:**

See weaknesses and questions

---

> ### Author Rebuttal · Authors · 2024-08-06
>
> ## Weaknesses
> 1. Thank you for your suggestion. We have indeed demonstrated the uniqueness of critical points under certain assumptions (Theorem 2 in the main text). Under these conditions, it can also be theoretically proven that if the gradient flow algorithm converges, it will drive the parameters to this unique critical point [1][2]. We will clarify this in the updated version. The numerical results presented in Section 3.1 (Figure 1 with sufficient training samples) further confirm the correspondence between the theoretically identified critical point and the numerically converging one.
> For more general cases, such as MNIST, Figure 5 in Appendix B shows that the post-training results are consistent with the theoretically identified critical point established in Theorem 5. For deeper networks and more complex datasets like CIFAR-10 and ImageNet, computing the critical points analytically becomes challenging.
>
> 2. Thank you for your valuable feedback. In Table 2 of the main text, we compare our Dit-CNNs with other quadratic methods. Our models show improvements in accuracy while utilizing only about one-third of the number of trainable parameters, highlighting its efficiency.
> Additionally, following your suggestion, we visualized the test accuracy in Figure 2.A of the rebuttal materials, which demonstrates that our model consistently improves in accuracy as the model size increases, while other quadratic methods tend to saturate. This indicates superior scaling properties for our approach.
>
> [1] Quarteroni, Alfio, Riccardo Sacco, and Fausto Saleri. Numerical mathematics. Vol. 37. Springer Science & Business Media, 2006.
>
> [2] Ambrosio, Luigi, Nicola Gigli, and Giuseppe Savaré. Gradient flows: in metric spaces and in the space of probability measures. Springer Science & Business Media, 2008.
>
> ## Questions
> Thank you for your insightful question. There are two possible reasons accounting for the difference: architecture and dataset (i.e., ResNet for CIFAR and ConvNeXt for ImageNet). To investigate the reason, we have trained Dit-ResNet on ImageNet-1K. After training, when we omitted the quadratic terms, the test accuracy dropped to 0.1%, which aligns with the Dit-ResNet results on CIFAR-10. So the key factor contributing to the differing performance drop levels in the ablation study is the use of entirely different network architectures for CIFAR and ImageNet.

---

### Author Rebuttal · Authors · 2024-08-06

Thank you to all the reviewers for their high-quality reviews. To address the reviewers' concerns, we have added more experiments and provided additional details about the model. We hope that our responses effectively address the reviewers' feedback .

 We have provided a comprehensive biological interpretation of our Dit-CNN, as illustrated in Figure 1.C of the rebuttal materials. Our Dit-CNN is inspired by neural networks in the visual system. For example, different types of cone cells encode various color (channel) information, and retinal ganglion cells receive inputs from multiple types of cone cells [1], the responses can be modeled as having receptive fields (convolutional kernels) related to different color channels ($w_1 *x_1, w_2 *x_2, w_3 *x_3 $). When multiple channel inputs are present, traditional CNNs simply linearly sum the corresponding responses. In contrast, neurons integrate these inputs with an additional quadratic term based on the dendritic bilinear integration rule. This approach leads to the formulation of our Dit-CNN after simplification. We believe this integration reflects a more biologically plausible mechanism compared to conventional methods.

[1] Kandel, Eric R., et al., eds. Principles of neural science. Vol. 4. New York: McGraw-hill, 2000.

---

### Decision · Program_Chairs · 2024-09-25

**Decision:**

Accept (poster)

**Comment:**

While the focus in the field has largely centered around building ever more complex and larger architectures of simple individual units, there is a lot of complexity and computation happening at the level of individual neurons in biological circuits. This paper focuses on a relatively understudied issue of where/how to incorporate some of the biophysical apparatus of individual neurons into neural networks. In this sense, I think that this paper contributes to the diversity of efforts in the field and it is useful to stimulate different approaches.

That said, I read the paper and responses and the paper is not very clearly written. The paper could benefit from editing by native speakers or further attention to grammar, prose, and exposition clarity.

The authors sent a note to the Area Chairs, expressing concern about reviewer opWC. I read the exchange of author responses and reviewer questions. While it is true that the reviewer only succinctly responded to the authors’ response, I do not see any inappropriate action from the reviewer. It is clear to me that opWC read the paper and provided useful critical feedback. Could opWC engage in a longer discussion of each of the rebuttal points? Probably. But overall, I feel that this is a rather reasonable review.

In all, I share many of the positive points noted by reviewers and I think that this work is different from others and deserves to be accepted, even though the overall discussion is borderline.